# Identification Method of Cotton Leaf Diseases Based on Bilinear Coordinate Attention Enhancement Module

**Mingyue Shao** [1,2]**, Peitong He** [1,2]**, Yanqi Zhang** [1,2]**, Shuo Zhou** [1,2]**, Ning Zhang** [1,2] **and Jianhua Zhang** [1,2,3,*]

1   Agricultural Information Institute, Chinese Academy of Agricultural Sciences, Beijing 100081, China
2   Key Laboratory of Agricultural Big Data, Ministry of Agriculture and Rural Affairs, Beijing 100081, China
3   National Institute of Nanfan, Chinese Academy of Agricultural Sciences, Sanya 572024, China
*   Correspondence: zhangjianhua@caas.cn

**Abstract:** Cotton is an important cash crop. Cotton diseases have a considerable adverse influence on cotton yield and quality. Timely and accurate identification of cotton disease types is important. The accuracy of cotton leaf disease identification is limited by unpredictable factors in natural settings, such as the presence of a complex background. Therefore, this paper proposes a cotton leaf disease identification model based on a bilinear coordinate attention enhancement module. It reduces the loss of feature information by bilinear coordinate attention embedding feature maps spatial coordinate information and feature fusion. Hence the model is more focused on the leaf disease region and reduces the attention to redundant information such as healthy regions. It also achieves the precise localization and amplification of attention to the leaf disease region through data enhancement, which effectively improves the recognition accuracy of cotton leaf diseases in a natural setting. By experiments, the identification accuracy of the proposed model is 96.61% and the parameter size is $21.55 \times 10^6$. Compared with other existing models, the identification accuracy of the proposed model is greatly improved without increasing the parameter size. This study can not only provide decision support for the timely diagnosis and prevention of cotton leaf diseases but also validate a paradigm for the identification of other crop leaf diseases.

**Keywords:** natural environment; identification of cotton leaf diseases; bilinear coordinate attention mechanism; data enhancement; ResNet





## 1. Introduction

Cotton is one of the main crops in the world, and is grown widely in India, China, the United States, and other countries [1]. It is the major source of income for local farmers. In the process of planting, cotton often suffers from infections, resulting in the decline of cotton yield and quality, and even no harvest in a large region in serious cases [2,3]. Therefore, cotton leaf disease identification methods are particularly important to provide decision support for the precision prevention of cotton leaf disease. Traditional leaf disease identification mainly relies on human labor, which requires experts with rich experience and knowledge to judge the type of leaf disease. That workflow not only takes time and effort but also has strong subjectivity and a high risk of misjudgment [4]. Laboratory testing methods are time-consuming and costly [5]. In order to overcome these shortcomings, automatic leaf disease identification based on computer vision has been highly valued by researchers.

In recent years, as the latest development direction of computer vision, the convolutional neural network has been widely used in crop leaf disease identification. For example, crop leaf disease identification models with different depths and a multi-scale feature information extraction strategy have been studied. For crop leaf disease images collected indoors under simple background conditions, the model can often achieve high identification accuracy. Sun et al. [6] used the improved AlexNet to identify 21,917 images of 14 species plants in PlantVillage, and the average test identification accuracy reached

99.56%. Mohanty et al. [7] took 54,306 images in PlantVillage as a baseline dataset and compared the performance of AlexNet and GoogleNet networks and concluded that the average identification accuracy of GoogleNet was better, up to 99.34%. Abbas et al. [8] used the deep neural network DenseNet121 model to identify tomato disease in PlantVillage images, with an accuracy of not less than 97.11%. Chellapandi et al. [9] also used the DenseNet network model to classify 14 plant species in PlantVillage and proved 99% identification accuracy. By fine-tuning the deep convolutional neural network DenseNet, Edna et al. [10] classified plant disease images in PlantVillage and achieved a test accuracy of 99.75%. Kamal et al. [11], Atila et al. [12] and Saleem et al. [13] also proposed to use of a lightweight network with fewer parameters to identify plant diseases in a PlantVillage dataset, and the final average classification accuracy was more than 98.34%. However, when the deep neural network model is applied in the actual production environment, the identification accuracy of the disease will be greatly reduced [14].

With the development of Deep Convolutional Neural Networks (DCNN), the research community has been gradually promoting studies from indoor scenes to complex natural environments [15–19]. Zhao et al. [20] identified potato disease images taken in the field based on a self-built deep neural network model but with an accuracy of only 87%. Zhang et al. [21] used the improved VGG16 to identify cotton diseases in the natural environment in the field; however, the accuracy was only 89.51%. Atole et al. [22] and Chen et al. [23] used a deep neural network model to identify rice disease images under complex background conditions but the highest accuracy was only 92%. Ramcharan et al. [24] identified cassava disease images taken in the field with an accuracy of 93%. Xing et al. [25] used the DenseNet model with deeper depth to identify 12,561 citrus disease images and obtained an accuracy rate of 93.42%. A large number of studies have proved that deep convolutional neural network models can effectively identify plant diseases, and can obtain very good identification results in the presence of a simple background. However, when the disease is in a complex background, the deep learning model will pay attention to many irrelevant features due to the influence of small disease spots, occlusion, illumination, and other factors, and pay less attention to the features of the disease in the region of interest [4].

Attention mechanisms can effectively compensate for the shortcomings of the DCNN model. Unlike deep learning models, attention mechanisms shift the focus from the image as a whole to the local region of interest, suppressing unnecessary features [26]; thus, they can better focus on the detailed information of the region of interest. Shang et al. [27] proposed a hybrid attention mechanism deep residual network, fusing spatial attention and channel attention modules on the basis of Resnet50 to classify 60 types of diseases of 10 crops in the PlantVillage datasets and proved that the network with a fused attention mechanism could obtain better recognition results. Li et al. [28] improved the ResNet18, ResNet34, and VGG16 models by incorporating an attention mechanism in order to classify apple diseases, and finally, ResNet18 obtained an accuracy of 98.5%. Zuo et al. [29] proposed an attention-based lightweight residual network to identify plant diseases in PlantVillage, and the final average accuracy was 98.89%. Yu et al. [30] used a deep model and an attention mechanism to identify apple leaf spot disease. They designed two sub-networks, a feature segmentation subnetwork for separating the disease from other regions with no relevant information such as the background, and a classification subnetwork for improving the classification accuracy of the disease spots. Eventually, the two sub-networks were fused and trained. The results demonstrate that this approach works better than state-of-the-art deep learning models. By considering the above research results, it can be seen that the recognition accuracy of the model after introducing an attention mechanism is higher than that of the model without introducing an attention mechanism. However, most of the attention mechanisms currently applied to disease recognition are for local information such as channel information or spatial information and lack attention to local information and coordinate information of diseases under a complex natural environment, etc.

In summary, the following problems still exist: (1) Relevant research is still more based on disease image recognition in the presence of a simple background; (2) when

it comes to features of the natural environment, such as uneven lighting and complex backgrounds, deep convolutional neural networks are less effective in recognizing disease types; and (3) current attention mechanisms lack a description of location information, leading to the loss of some feature information. Therefore, the purpose of this study was to, as far as possible, try to overcome the above problems. We use an attention mechanism to transform the focus from the whole image to the disease region, which suppresses unnecessary features and better focuses on the detailed information of the region of interest [31]. Specifically, we construct an identification model of cotton leaf diseases based on a bilinear coordinate attention enhancement mechanism (BCAEM). The model uses the ResNet34 as the feature extraction backbone and combines the bilinear model theory and the data enhancement method guided by an attention mechanism, which can extract more fine-grained feature points.

The main highlights of this article are as follows:

1.　The bilinear coordinate attention mechanism pays more attention to the lesion features.
2.　Coordinate-aware feature fusion improves the accuracy of disease area localization.
3.　The attention-guided data enhancement model can learn more discriminative features.
4.　The proposed model achieves higher accuracy with fewer parameters.

## 2. Materials and Methods

### 2.1. Materials

#### 2.1.1. Original Images Acquisition

In this paper, images of eight types of cotton leaf diseases—Anthracnose (*Colletotrichum gossypii* Southw.), Bacterial blight (*Xanthomonas campestris* pv. *malvacearum* (Smith) Dye), Brown spot (*Phyllosticta gossypina* El.et Mart.), Fusarium wilt (*Fusarium oxysporum* f. sp. *vesinfectum* (Atk.) Snyder et Hansen), Leaf curl (*Gossypium hirsutum* L.), Red leaf, Ring spot (*Alternaria macrospora.*), and Verticillium wilt (*Verticillium dahliae* kieb.)—and one healthy type of cotton are collected. This covers the main types of leaf diseases occurring in the cotton areas of China. The cotton fields were planted at the Langfang Scientific Research Pilot Base of the Chinese Academy of Agricultural Sciences (N: 39°27′55.59″, E: 116°45′28.54″) from 2019 to 2021 for the purpose of collecting cotton leaf disease images in a natural setting. To acquire cotton images under various natural environmental conditions, images are collected in the morning, noon, and evening, on sunny and cloudy days, and at a distance of 20–50 cm from the target cotton. Images are taken using a Canon EOS 850D SLR digital camera(Canon Inc., Body: Japan; Lens: Taiwan, China), and all images are acquired in automatic exposure mode and saved in JPG format. A cotton leaf disease dataset of 5903 images in the natural environment has been built—816 images of cotton anthracnose, 503 images of cotton bacterial blight, 916 images of cotton brown spot, 537 images of cotton fusarium wilt, 499 images of cotton leaf curl disease, 763 images of cotton red leaf, 503 images of cotton ring spot, 856 images of cotton verticillium wilt, and 510 images of healthy leaves. The types of these cotton leaf disease images were identified by two professional cotton pathologists. The types, figures, image samples, and key features of each leaf disease in the dataset are presented in Table 1.

**Table 1.** The types, figures, image samples, and key features of each leaf disease in the dataset.

| Type of Disease | Figures | Image | Key Features |
|---|---|---|---|
| Cotton anthracnose | 816 |  | On the edge of the raw edge semicircle brown disease spots, reddish brown, after drying off the cripple cotyledon edge. |
| Cotton bacterial blight | 503 |  | The initial stage is oil-stain-shaped spots, the later stage features polygonal or irregular spots. |
| Cotton brown spot | 916 |  | Early on the edge of the cotyledon or other parts of the tip, small purple dots; these expand into the middle and turn brown with a purple edge, and change from circular to an irregular shape. |
| Cotton fusarium wilt | 537 |  | Cotyledons or leaf apex from the tip area begin to turn yellow; this discoloration gradually expands inside and finally causes leaf loss. |
| Cotton leaf curl | 499 |  | At the beginning of the disease, the top of the tender leaves is slightly curled, the curl is then aggravated and the leaf abaxial surface articulates. |
| Cotton red leaf | 763 |  | In the early stage of the disease, all areas except the veins and their vicinity, which remain green, turn purple-red or reddish brown. |
| Cotton ring spot | 503 |  | The early onset of brown spots, accompanied by a purple halo, followed by gray-brown near-circular spots; spots on both sides have concentric rings. |
| Cotton verticillium wilt | 856 |  | Pale yellow patches appeared between the leaf margin and veins, then gradually expanded, and the leaves lost their green color; the main vein and its surroundings remain green, and disease leaves appear palmate mottled. |
| Healthy | 510 |  | None. |

### 2.1.2. Construction of Cotton Leaf Disease Dataset

In order to enhance the learning effect of the Convolutional Neural Networks (CNN) model and reduce the problem of over-fitting and under-fitting of the model, the dataset is pre-processed. Through blurring, flipping, brightening, darkening, random rotation, random noise, and other image enhancement methods, the image dataset is expanded to achieve data enhancement. An example of image enhancement is shown in Figure 1. The details of the final constructed cotton leaf disease dataset are shown in Table 2.

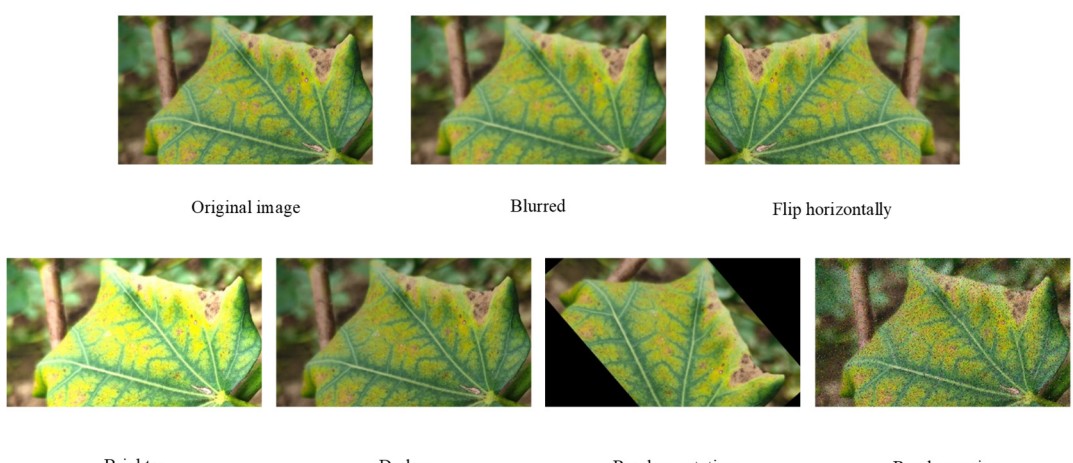

**Figure 1.** Examples of image enhancement.

**Table 2.** Details of the constructed cotton leaf disease dataset.

| Type of Disease | Figures | |
|---|---|---|
| | **Original Dataset** | **Expanded Dataset** |
| Cotton anthracnose | 816 | 2957 |
| Cotton bacterial blight | 503 | 2921 |
| Cotton brown spot | 916 | 4529 |
| Cotton fusarium wilt | 537 | 3120 |
| Cotton leaf curl | 499 | 2899 |
| Cotton red leaf | 763 | 4429 |
| Cotton ring spot | 503 | 2921 |
| Cotton verticillium wilt | 856 | 4958 |
| Healthy | 510 | 2964 |
| Total | 5903 | 31,698 |

### 2.2. Methods

In cotton fields in a natural setting, the region of the image containing cotton leaf disease is much smaller than the background and healthy leaf regions [32]. In addition, some cotton leaf diseases are very similar to each other. The deep convolutional neural network struggles to distinguish useful and noisy information in the network model formation process. This can result in bad identification performance [33]. For the reasons above, current traditional convolutional neural networks are not rich enough in the extraction of leaf disease spot features, especially local information, and thus have low accuracy for cotton leaf disease recognition in a natural setting. For example, cotton verticillium wilt and cotton fusarium wilt are both major leaf diseases that affect cotton and can occur throughout the cotton reproductive period and both have similar damage symptoms. At the beginning of the disease, in both cases, leaves appeared to fade to green and some areas turned yellow in the late stages of development, the two leaf diseases often cause the plant leaf to die. Therefore, to better identify cotton leaf diseases in a natural setting, a cotton leaf disease identification model based on the bilinear coordinate attention enhancement module (BCAEM) is developed in this paper.

### 2.2.1. ResNet

ResNet, a residual structure network proposed by He et al. [34], is a deep convolutional neural network architecture that solves the network degradation problem by stacking residual structures composed of shortcut connections and identity mapping. In this paper, ResNet is used as the backbone network to extract the features of cotton leaf diseases. In the residual structure network, at the $n$-th residual structure, $x_n$ is taken as the input, and $x_{n+1}$ is taken as the output. The residual can be expressed as:

$$y_n = h(x_n) + F(x_n, W_n), \tag{1}$$

$$x_{n+1} = f(y_n), \tag{2}$$

where $F(x_n, W_n)$ stands for the learned residuals and $f(y_n)$ stands for the activation function. Since the residuals are generally smaller, residual learning requires a little less learning, so residual learning is easier. So, when residuals exist, if $h(x_n)$ denotes the constant mapping, then

$$x_{n+1} = x_n + F(x_n, W_n), \tag{3}$$

$x_{n+1}$ is easy to fit. Even if there is no residual (as shown in Equation (4)), the network performance will not degrade.

$$x_{n+1} = x_n, \tag{4}$$

To reduce the number of network parameters and computational volume, yet ensure the effective extraction of sample features, we choose ResNet34 as the backbone network of the model.

### 2.2.2. Bilinear Coordinate Attention Mechanism

Coordinate attention is an efficient attention mechanism proposed by Hou et al. [35]. However, the average pooling of the traditional coordinate attention mechanism losses features information and thus affects the network's ability to identify objects. In this paper, the traditional mechanism for coordinating attention has been improved. A bilinear coordinate attention mechanism (BCAM) is proposed to solve the above issue. It not only allows effective integration by spatial coordinate information and attention but also reduces the loss of feature information by means of feature fusion. This makes the model more focused on the diseased region and reduces the attention to the background, healthy region, and other redundant information. Figure 2 shows the structure of the bilinear coordinate attention mechanism (BCAM).

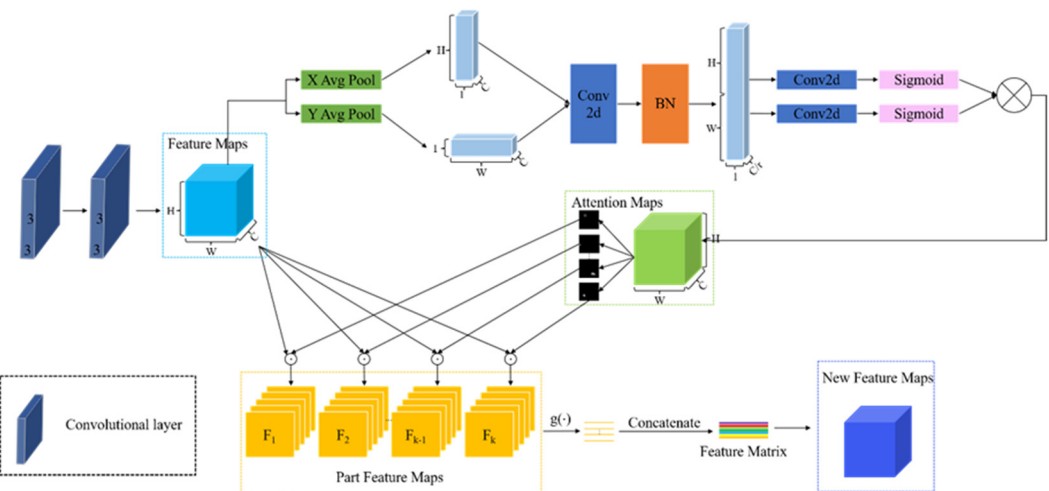

**Figure 2.** Dual bilinear attention module—DBAM, where ⊙ denotes the element-by-element multiplication of two tensors, $g(\cdot)$ is a feature extraction function, $W$, $H$, and $C$ denote width, height, and channel size.

The bilinear coordinate attention mechanism can be divided into three parts—location information embedding, coordinate attention generation, and bilinear coordinate attention generation. As shown in Figure 2, after the convolutional processing of the residual blocks, a series of features, designated as *F*, is provided to the output Feature Maps, where $F \in R^{W \times H \times C}$, and *W*, *H*, and *C* denote width, height, and channel size, respectively.

By dividing the feature factors into pairs of one-dimensional features, Equations (5) and (6) are able to determine the location information of the features. To be more precise, $x_c$ is the intermediate eigenvector representation of *F* on the *c*-th channel. The pooling operations are carried out on a given input $x_c$ in the same channel dimension from the height (*h*) and width (*w*) dimensions, respectively, to produce two one-dimensional feature vectors ($z_c^h$ and $z_c^w$) for various directions of the same feature. In one direction, dependencies are established, while in the other, the features' locations are maintained.

$$z_c^h = \frac{1}{W} \sum_{0 \le i \le W} x_c(h, i),\tag{5}$$

$$z_c^w = \frac{1}{H} \sum_{0 \le j \le H} x_c(j, w),\tag{6}$$

where $z_c^h$ denotes the output of the *c*-th channel at height *h*, $z_c^w$ denotes the output of the *c*-th channel over the width *w*, and $x_c$ denotes intermediate eigenvectors on the c-th channel.

Two one-dimensional feature vectors, $z_c^h$ and $z_c^w$ are fully employed to build a coordinate attention mechanism to accomplish information exchange between channels. The two feature vectors $z_c^h$ and $z_c^w$ are primarily connected using Equation (7) before being placed into a $1 \times 1$ convolution to produce the attention map.

$$f = \delta\left(F_1\left(\left[z_c^h, z_c^w\right]\right)\right),\tag{7}$$

where $[z_c^h, z_c^w]$ signifies the concatenate operation in the spatial dimension, $F_1$ is the convolutional transform function, $f \in R^{(W+H) \times C/r}$ is the intermediate feature vector, and $\delta$ is the nonlinear activation function.

Equations (8) and (9) serve mainly to reduce the dimensionality to lower model computation. Specifically, two $1 \times 1$ convolutional transforms and nonlinear activation functions are used to reduce the computational overhead by again decomposing f into two tensors with the same number of channels as the input $x_c$ and expanding the two tensors into attention weights.

$$g_h = \sigma(F_h(f_h)),\tag{8}$$

$$g_w = \sigma(F_w(f_w)),\tag{9}$$

where $g_h$ and $g_w$ are the generated attention weights, $\sigma$ is the sigmoid function, $F_h$ is the convolutional transform function in the horizontal direction, and $F_w$ is the convolutional transform function in the vertical direction.

The Attention Maps with embedded location information, which is designated as A, are produced by multiplying the two attention weights produced in Equations (8) and (9) with the input characteristics, as illustrated in Equation (10).

$$y(i, j) = x(i, j) \times g_h(i) \times g_w(j),\tag{10}$$

Equation (11) achieves feature fusion between local feature maps and reduces the loss of feature information. The specific calculation process is as follows: each attention map *A* is divided into *k* parts, and then the feature map *F* is multiplied with it element-wise

to obtain the $k$ attention-guided local feature maps $f_k$, where $f_k \in R^{1 \times N}$. Finally, the local features are stitched together and output as the final feature matrix.

$$P = \Gamma(A, F) = \begin{pmatrix} g(A_1 \odot F) \\ g(A_2 \odot F) \\ \dots \\ g(A_k \odot F) \end{pmatrix} = \begin{pmatrix} f_1 \\ f_2 \\ \dots \\ f_k \end{pmatrix},$$

(11)

where $P \in R^{M \times N}$, $\Gamma(A, F)$ denotes the bilinear attention operation, $g(A_k \odot F)$ denotes the pooling operation, $A_k$ denotes the $k$-th part of the attention graph, and $f_k$ denotes the $k$-th attention-guided local feature map.

Figure 3 illustrates the effect of the bilinear coordinate attention mechanism (BCAM) on the extraction of diseased regions.

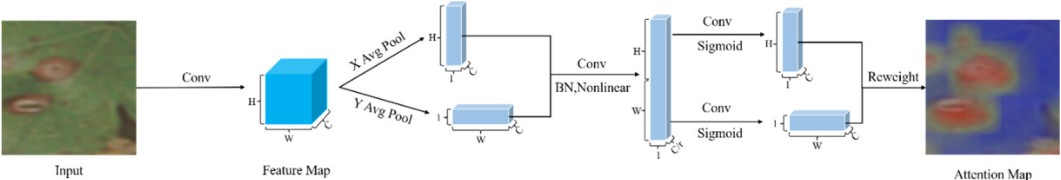

**Figure 3.** The effect of the bilinear coordinate attention module on the extraction of the disease region, where *W*, *H*, and *C* denote width, height, and channel size.

### 2.2.3. Bilinear Coordinate Attention-Guided Data Enhancement

In this paper, a data enhancement method guided by bilinear coordinate attention is adopted. Based on the bilinear coordinate attention map, the local features are enhanced by attention cropping and attention dropping, and the model is encouraged to extract features from multiple discriminative parts to realize the accurate localization and amplification of the lesion region, to improve the identification accuracy of the overall model.

The acquisition of the cropped attention image is as follows: a coordinate attention map $A_k$ is randomly selected and a threshold $\theta_c \in [0, 1]$ is set. If $A_k (i, j) > \theta_c$, then the element is set to 1 and in other cases, it is set to 0. As shown in Equation (12). Then, we frame all the elements equal to 1 and cut and zoom to the size of the original image.

The acquisition of the dropped attention image is as follows: a coordinate attention map $A_k$ is randomly selected and a threshold $\theta_d \in [0, 1]$ is set. If $A_k (i, j) > \theta_d$, then the element is set to 0, and in other cases, it is set to 1. as shown in Equation (13). Then, we erase all the elements equal to 0 in the original image.

Finally, the image data generated by the two operations are fed into the convolutional neural network as the enhanced data for training. The process of data enhancement image generation guided by coordinate attention is shown in Figure 4.

$$C_k i, j = \begin{cases} 1, A_k(i, j) > \theta_c \\ 0, otherwise \end{cases},$$

(12)

$$D_k i, j = \begin{cases} 0, A_k(i, j) > \theta_d \\ 1, otherwise \end{cases},$$

(13)

### 2.2.4. Identification of Cotton Leaf Diseases Based on Bilinear Coordinate Attention Enhancement Module

The model consists of three parts—backbone network, bilinear coordinate attention mechanism, and data enhancement module guided by bilinear coordinate attention. ResNet34 is used as the backbone network to extract the feature information of the leaf disease image. The bilinear coordinate attention mechanism extracts feature information in the residual channel by embedding feature coordinate information and feature fusion. In addition, the long-term dependence relationship between convolutional channels is

established to solve the problems of inaccurate localization of regions of interest and loss of feature information found with other attention mechanisms. Finally, the feature of the region of interest is extracted. The coordinate-attention-guided data enhancement module solves the problem that common data enhancement methods bring a lot of noise information to small images. The model was guided by the coordinate attention map to crop and drop the diseased region so that the network could pay attention to the more fine-grained feature information of the diseased region. Therefore, more features of leaf disease parts of the image can be paid attention to improve the robustness of identification. The model structure is shown in Figure 5. Algorithm 1 shows the model algorithms used in this paper.

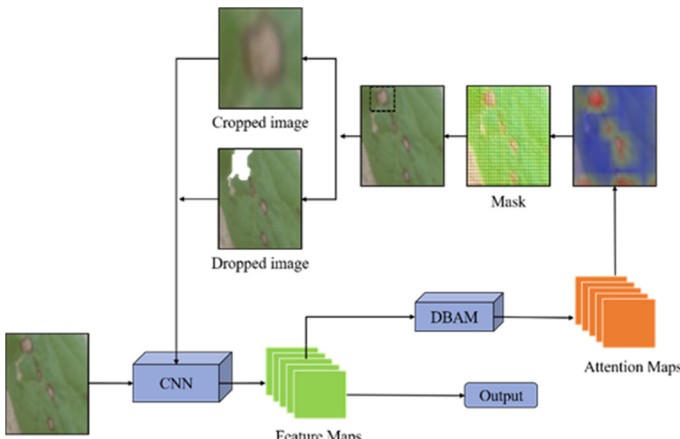

**Figure 4.** Coordinate attention-guided data enhancement image generation process, where CNN denotes Convolutional Neural Network, DBAM denotes dual bilinear attention module.

---

**Algorithm 1:** Algorithm for Model

---

Input:
Training images: Enhanced Image.
Output:
Predicted labels of the Test set.

1. Set batch size $\leftarrow$ 64, optimizer Stochastic Gradient Descent (SGD), learning rate $\leftarrow 1 \times 10^{-2}$, momentum $\leftarrow$ 0.9, weight decay $\leftarrow 1 \times 10^{-5}$; lr scheduler $\leftarrow$ StepLR: step size $\leftarrow$ 2, gamma $\leftarrow$ 0.9, image dimensions to 256;
2. For $i \leftarrow 1$ to $N$ do:

- Input the training images into the model loaded with pre-trained weights and obtain feature matrix, attention map, crop images, and drop images;
- Train the model;
- Loss backpropagation;
- Update the model parameters.

3. End for;
4. Compute the overall classification accuracy of the test dataset;
5. Classify test images.

---

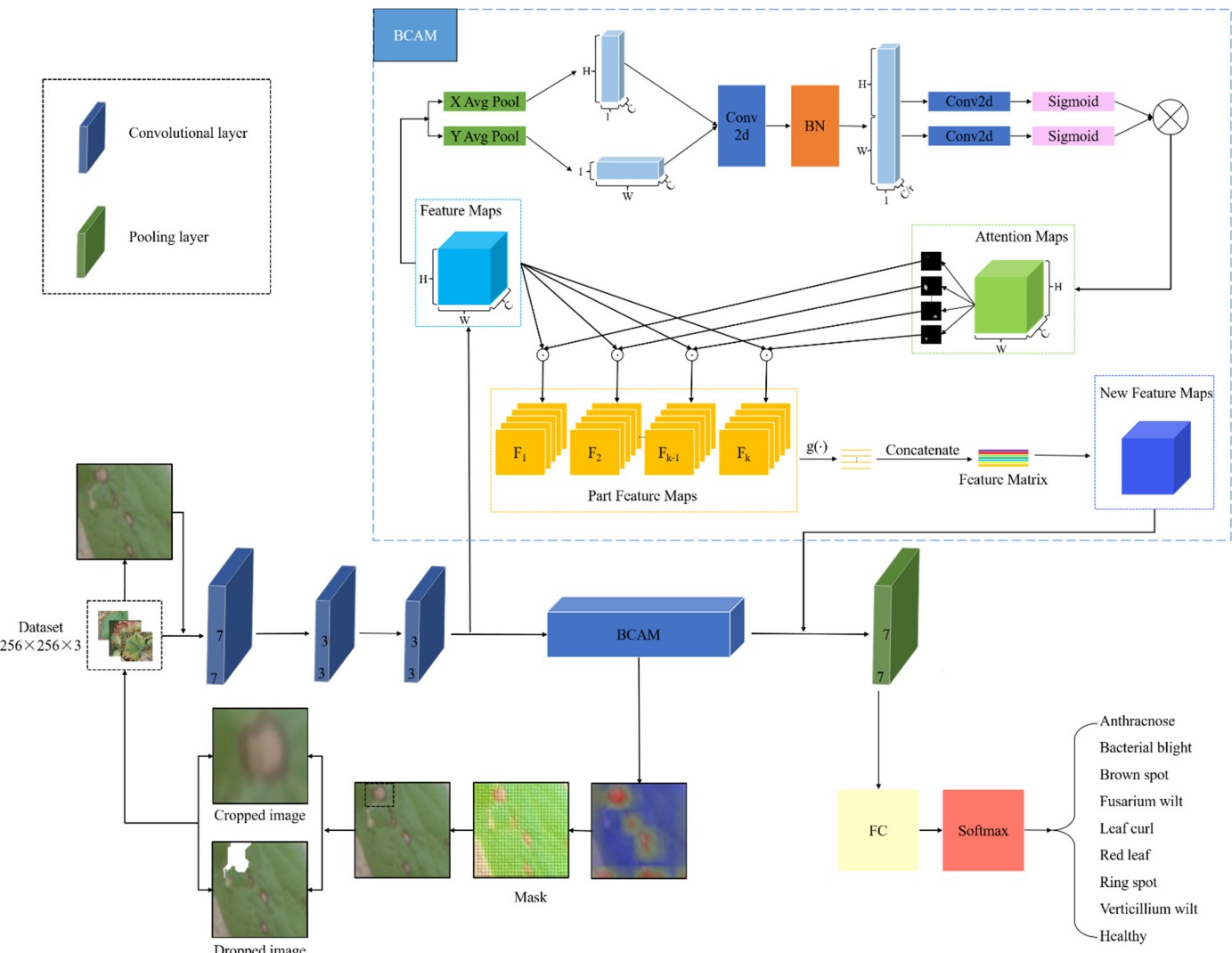

**Figure 5.** Identification model of cotton leaf disease based on bilinear coordinate attention-enhancement module, where $\odot$ denotes the element-by-element multiplication of two tensors, $g(\cdot)$ is a feature extraction function, *W*, *H*, and *C* denote width, height, and channel size, BCAM denotes bilinear coordinate attention mechanism.

## 3. Experimental Setting and Evaluation Metrics

### 3.1. Experiment Setting

Information about the test platform used for the experiment is as follows: Windows 10, Intel (R) Core (TM) i7-7700 CPU @ 3.60 GHz core processor, 32.0 GB of on-board RAM, 200 GB of disk capacity, NVIDIA TITAN Xp COLLECTORS EDITION graphics card to accelerate model training. The software environment is Python 3.6, PyTorch 1.7.1, CUDA 10.1.

In this experiment, the hyperparameters are specifically set as follows: training epoch is set to 100, the batch size is set to 64, Stochastic Gradient Descent (SGD) is used as the optimizer, momentum is set to 0.9, weights are decayed to $1 \times 10^{-5}$, and $1 \times 10^{-2}$ is used as the initial learning rate. The backbone network is loaded with weights pre-trained on the publicly available dataset PlantVillage, and the weights are continuously updated in all layers during the training process. After each epoch of training, the model is validated on the validation set while saving the best hyperparameters of the training model to the weights.

### 3.2. Evaluation Metrics

In this paper, the performance of the model in terms of accuracy, precision, recall, and specificity is evaluated, as shown in Equations (14)–(17), where TP denotes true positive, TN true negative, FP false positive, and FN false negative.

$$\text{Accuracy} = \frac{(\text{TP} + \text{TN})}{(\text{TP} + \text{TN} + \text{FP} + \text{FN})}, \tag{14}$$

$$\text{Precision} = \frac{\text{TP}}{\text{TP} + \text{FP}}, \tag{15}$$

$$\text{Recall} = \frac{\text{TP}}{\text{TP} + \text{FN}}, \tag{16}$$

$$\text{Specificity} = \frac{\text{TN}}{(\text{FP} + \text{TN})}, \tag{17}$$

## 4. Results and Discussion

The dataset created in Table 1 is used as the research baseline. This dataset is collected in a natural setting, where the images have different backgrounds, light and shadow conditions, and other noise factors, which is very challenging. The dataset is divided into training, validation, and test sets in the ratio 8:1:1, and only the training set is augmented by the data enhancement method mentioned in Section 2.1.2. Ultimately, there is a total of 31,698 images, corresponding to 30,584 in the training set, 560 in the validation set, and 554 in the test set (as shown in Table 2). The following experiments are carried out on the augmented image dataset, and all the results are the average identification accuracy obtained after five repeated training and testing cycles.

### 4.1. Ablation Study

In order to clarify the contribution of each module in the model on the performance of the model, model ablation experiments are carried out. ResNet34 is used as the backbone network to integrate three modules—Coordinated Attention (CA), BCAM, and BCADE—to verify the impact of the combination of different modules on the performance of the model, to optimize the combination of modules, and construct the optimal model. The ablation test results of the model are shown in Table 3.

**Table 3.** Ablation test results of the constructed model.

| Number | ResNet34 | +CA | +BCAM | +BCADE | Accuracy/% | Parameters/M |
|:---:|:---:|:---:|:---:|:---:|:---:|:---:|
| 1 | ✓ | | | | 95.18 | 21.29 |
| 2 | ✓ | ✓ | | | 95.36 | 21.39 |
| 3 | ✓ | ✓ | | ✓ | 95.89 | 21.46 |
| 4 | ✓ | | ✓ | | 96.43 | 21.55 |
| 5 | ✓ | | ✓ | ✓ | 96.61 | 21.55 |

where CA stands for Coordinated Attention, BCAM stands for Bilinear Coordinate Attention Mechanism, and BCADE stands for Bilinear Coordinated Attention En-hancement Module.

In Table 3, (1) uses only ResNet34 for cotton leaf disease identification, and the accuracy is only 95.18% and the number of parameters is $21.29 \times 10^6$. (2) is the combination of ResNet34 + CA. Based on ResNet34, coordinate attention is added to the residual block, and the accuracy is 95.36%, which is 0.18% higher than that of ResNet34 alone because the coordinate attention mechanism could effectively improve the feature extraction ability of the model. The number of parameters of this model is $21.39 \times 10^6$, which is $0.1 \times 10^6$ higher than that of ResNet34 alone. (3) is the method of ResNet34 + CA + BCADE. The data enhancement guided by coordinate attention enriches the dataset, therefore, an identification accuracy of 95.89% is obtained, which is 0.53% higher than that of ResNet34 + CA. (4) is the method of ResNet34 + BCAM. This method uses a bilinear coordinate

attention mechanism, which improves the accuracy by 1.07% compared to ResNet34 + CA's leaf disease identification model based on a single linear coordinate attention. This is because the use of a bilinear coordinate attention structure can better fuse the multi-level features and reduce the loss of features. (5) is the method of ResNet34 + BCAM + BCADE. This method obtains 96.61% identification accuracy and has $21.55 \times 10^6$ parameters.

In terms of accuracy, the identification accuracy of the model constructed by the combination of ResNet34 + BCAM + BCADE is 1.4% higher than the identification accuracy of the original ResNet34 alone. Additionally, by comparing the results of (4) and (5), it can be seen that the addition of the BCADE module increases the accuracy by only 0.18%. Therefore, it can be concluded that the BCAM module has the largest contribution in terms of accuracy improvement. In terms of the number of parameters, the number of parameters of ResNet34 + BCAM + BCADE is only 0.29 M more than that of ResNet34 alone. In conclusion, the ablation test shows that ResNet34 + BCAM + BCADE module had the best performance compared with other models. Compared with ResNet34 alone, the identification accuracy is improved. Therefore, a cotton leaf disease identification model based on a bilinear coordinate attention enhancement module is validated.

Figure 6 displays the identification results of the proposed model for each of the nine leaf disease types. As shown in the figure, the proposed model has good results in identifying cotton leaf curl, cotton ring spot, and healthy leaves, and can also successfully identify the other six types. It can be summarized from the figure that the model proposed in this paper is effective in identifying nine cotton leaf diseases; however, some images of cotton anthracnose, cotton brown spot, cotton fusarium wilt, and cotton verticillium wilt are misidentified. This is because the leaf disease characteristics of cotton anthracnose and cotton brown spot, cotton fusarium wilt, and cotton verticillium wilt are similar and not easy to distinguish.

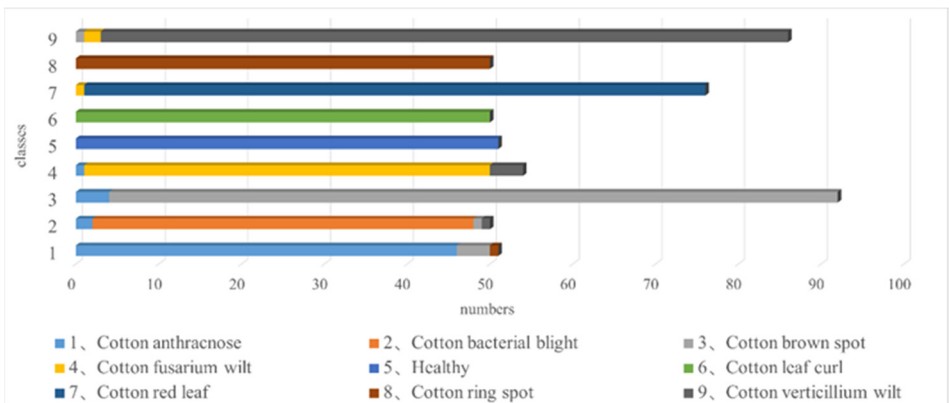

**Figure 6.** Classification results of the proposed model for nine leaf disease types.

In this paper, the visualized heat maps of model attention are presented from three aspects, including different leaf diseases, different backgrounds and light situations, and different noise disturbances. The attention concentration of the model proposed in this paper on the disease region is tested, as shown in Figure 7, where red color indicates the highest contribution and blue color indicates the lowest contribution. From Figure 7a, it can be seen that the proposed model can accurately locate irregular and striped spots as well as large and small spots. As can be seen in Figure 7b, when there are complex backgrounds, shadows, and reflections in the leaf disease images, the model is easily influenced, and the region of concern includes part of the background region in addition to the disease region. Illumination such as shadows and reflections have less impact on the model than leaf disease images with complex backgrounds and the main region of interest for the model is the disease region. From Figure 7c, it can be seen that, when artificial operations such as random noise and blurring are used to simulate real photographic situations, the model is almost unaffected by the interference information and locates the disease spots

accurately, and the focus region of the model is consistent with the that of human eyes. Therefore, it is proved that the proposed model in this paper has a good identification performance on cotton leaf diseases for different disease spot morphology, light effects, and interference information.

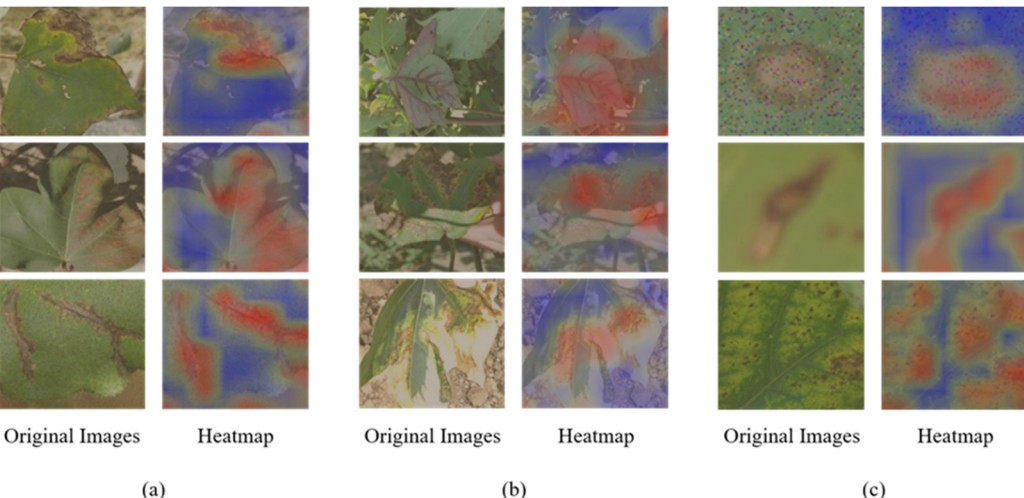

**Figure 7.** Visualization results of feature images of test sets, where (**a**) represents heatmaps for different leaf diseases, (**b**) represents heatmaps for different backgrounds and lighting, and (**c**) represents heatmaps with different noise interference.

*4.2. Comparative Evaluation*

In order to further validate the superiority of the proposed model, a comparative evaluation is carried out under the same experimental conditions. Based on the cotton leaf disease datasets, including a training set of 30,584 images, a validation set of 560 images, and a test set of 554 images. The model proposed in this paper is compared with AlexNet, VGG16, GoogleNet, ResNet34, ResNet50, ResNet101, SENet, and CBAM from five aspects: accuracy, precision, recall, specificity, and the number of parameters.

To verify the effect of the bilinear attention module proposed in this paper on the model performance, comparison experiments based on different attention mechanisms are designed. Figure 8 shows heat maps of cotton leaf disease identification for each of the three attention mechanisms. The red region is the critical region considered by the model and the model mainly extracts the features in this part, while the blue part is considered to be redundant features. It can be observed that, although CBAM is accurate for cotton leaf disease identification, it cannot locate the disease region well. This is probably because it is more susceptible to interference from redundant features such as the background. SENet is more accurate in locating the disease region parts compared with CBAM. However, the proposed model can better help locate the region of interest. The proposed model focuses more on the features of the disease region, which avoids the interference of leaf disease-independent information such as cotton leaf features and image background and improves the generalization ability of the model for plant leaf disease identification which suggests that using only GAP branches or GMP branches may lead to the problem of feature loss or inaccurate localization. In summary, the model designed in this paper can identify cotton leaf diseases well in a natural setting and is robust to real-world noise such as background, shading, and lighting.

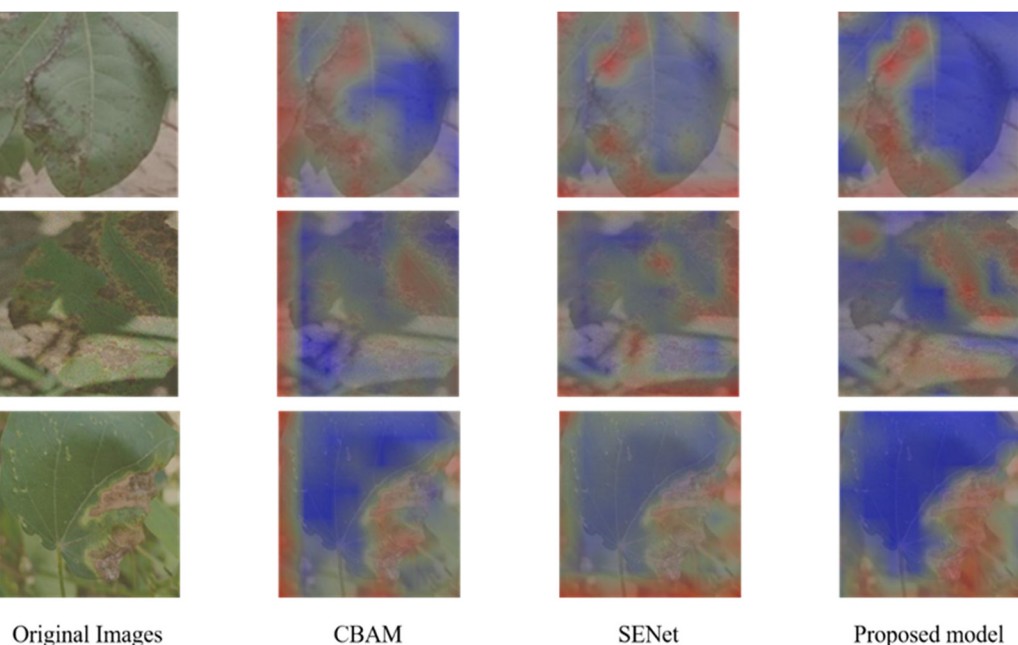

| Original Images | CBAM | SENet | Proposed model |

**Figure 8.** Visual results of cotton leaf disease identification based on SENet, CBAM, and the proposed model.

The results of the tests by the models on the cotton leaf disease dataset are shown in Table 4. As shown in the table, in terms of accuracy, that of VGG16 and VGG19 is below 90%; the identification accuracy of AlexNet, GoogleNet, ResNet34, ResNet50, ResNet101, and SENet is between 90% and 96%. And the identification accuracy of both CBAM and the proposed model is above 96%, with the proposed model having the highest identification accuracy of 96.61%, which is 0.54%~10.68% higher than the classical network. In terms of precision, the precision values of VGG16 and VGG19 are below 0.900; AlexNet, GoogleNet, ResNet34, ResNet50, ResNet101, SENet, and CBAM are between 0.900 and 0.960. And the precision values of the proposed model are above 0.960, which is higher than that of the classical networks by 0.012 to 0.093. In terms of recall, the recall of both VGG16 and VGG19 is below 0.900; the recall of AlexNet, GoogleNet, ResNet50, ResNet101, SENet, and CBAM is between 0.900 and 0.950; and the recall of ResNet34 and the proposed model are higher, both being above 0.950. The proposed model has the highest recall of 0.960, which is higher than the classical networks by 0.007 to 0.093. In terms of specificity, the specificity of AlexNet, VGG16, and VGG19 are all below 0.990; and the specificity of GoogleNet, ResNet34, ResNet50, ResNet101, SENet, CBAM, and the proposed model are all above 0.990, with the proposed model having the highest specificity of 0.995, which is higher than the other models by up to 0.011. In terms of the number of parameters, VGG16, VGG19, and ResNet101 require a larger number of parameters for training, all higher than $40 \times 10^6$; AlexNet, GoogleNet, ResNet34, ResNet50, SENet, CBAM, and the proposed model require fewer parameters, between $10 \times 10^6$ and $25 \times 10^6$, which can achieve faster training. The comprehensive analysis shows that the proposed model has the best recognition performance in terms of accuracy, precision, recall, and specificity, which can effectively improve the recognition rate of cotton disease leaves. Compared with traditional deep convolutional neural networks and traditional attention mechanisms, the proposed model obtains the best recognition results with the least increase in parameters. This is because the proposed model uses a bilinear coordinate attention mechanism, which can obtain more fine-grained information than the traditional deep convolutional neural network. Also, it possesses less loss of feature information than the traditional attention mechanism.

**Table 4.** Comparison test results between the constructed model and the other nine mainstream models.

| Models | Accuracy/% | Precision | Recall | Specificity | Parameters/M |
|--------|-----------|-----------|--------|-------------|--------------|
| AlexNet | 91.79 | 0.925 | 0.914 | 0.989 | 14.60 |
| VGG16 | 88.93 | 0.885 | 0.882 | 0.986 | 134.30 |
| VGG19 | 87.32 | 0.870 | 0.867 | 0.984 | 139.61 |
| GoogleNet | 94.46 | 0.944 | 0.945 | 0.993 | 10.33 |
| ResNet34 | 95.18 | 0.951 | 0.953 | 0.994 | 21.29 |
| ResNet50 | 94.82 | 0.933 | 0.931 | 0.992 | 23.53 |
| ResNet101 | 95.00 | 0.951 | 0.949 | 0.994 | 42.52 |
| SENet | 95.54 | 0.940 | 0.943 | 0.993 | 21.61 |
| CBAM | 96.07 | 0.927 | 0.930 | 0.991 | 21.61 |
| Proposed model | 96.61 | 0.963 | 0.960 | 0.995 | 21.55 |

To compare the specific identification results of different models on different leaf disease types, Figure 9 illustrates the confusion matrix plot for the 10 models. Each row represents the true label of each type, and each column is the prediction of the model. The value in the table is the probability that the row type is predicted to be the column type, and the diagonal value is the probability of correct classification. From the confusion matrix plot, it can be seen that VGG16 and VGG19 have a lower probability of correct prediction for eight leaf disease types and the one healthy class. VGG16 has a better prediction for cotton leaf curl, cotton red leaf, and cotton verticillium wilt, which are all above 0.9; while VGG19 has better results for cotton bacterial blight, cotton leaf curl, and healthy class identification with a correct probability above 0.9. AlexNet, GoogleNet, ResNet34, ResNet50, ResNet101, SENet, and CBAM has a higher probability of correct prediction for each class than VGG16 and VGG19 on average. In a comprehensive analysis, the most easily misidentified leaf disease types in the above models are cotton anthracnose, cotton brown spot, cotton ring spot, cotton fusarium wilt, and cotton verticillium wilt. Among them, judging by each value in the table, it can be found that the model often confuses three leaf diseases, cotton anthracnose, cotton brown spot, and cotton ring spot. Cotton anthracnose shows semicircular brown spots at the seedling stage, while cotton ring spot has circular brown spots at the seedling stage. The edges of the spots of cotton anthracnose, cotton brown spot, and cotton ring spot show purple color at the mature stage. In addition, the models can easily cause confusion between cotton fusarium wilt and cotton verticillium wilt. The main feature of the common leaf disease characteristics in the field for both leaf diseases is the fading of the leaves to yellow. Because of the similarity between leaf disease characteristics, the above models result in high error rates in the prediction of these leaf diseases. The proposed model can accurately identify cotton bacterial blight, cotton leaf curl, cotton red leaf, cotton ring spot, and healthy leaves, and the precision is about 1.0. The precision of cotton brown spot, cotton fusarium wilt, and cotton verticillium wilt identification is 0.94, which represents relatively accurate identification. The identification precision of cotton anthracnose is the worst, only 0.87. It is observed that 8% of cotton anthracnose is considered cotton brown spot by the model. Overall, compared to other models, the proposed model has less probability of misjudgment for each type of leaf disease and a stronger ability to extract subtle features.

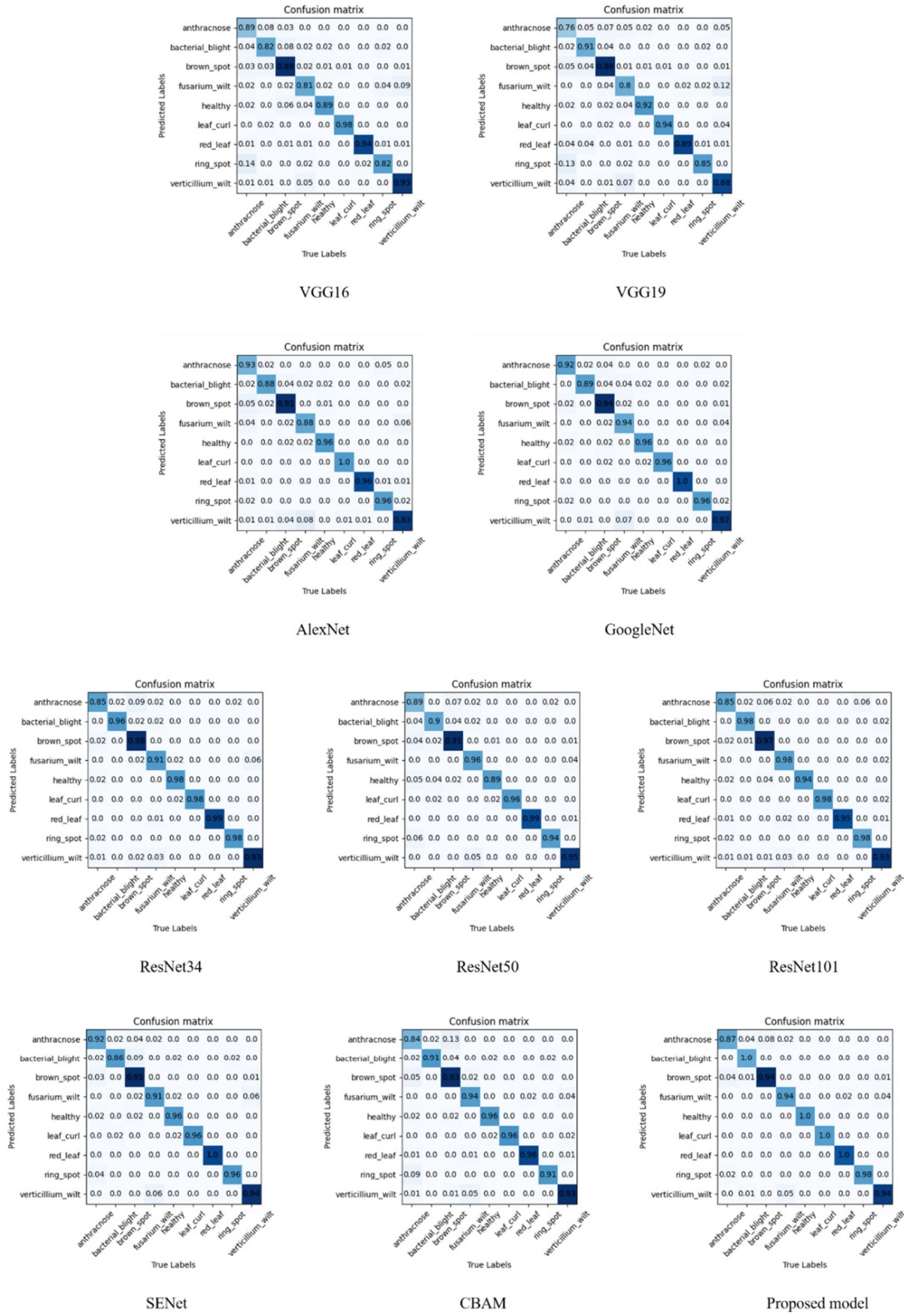

**Figure 9.** Confusion matrix. Where each row represents the true label of each type, each column is the predicted label of the model, the median value is the probability that the row type is predicted to be a column type, and the diagonal value is the probability of correct classification.

## 5. Conclusions

This work presents a novel cotton leaf disease identification model based on a bilinear coordinate attention enhancement module built on a ResNet34 backbone. Unlike traditional deep convolutional neural network models, the proposed model overcomes the issue of low recognition accuracy caused by external factors such as uneven illumination. It can locate the disease region more accurately, discard the redundant information in the image, extract the key region features, and realize the high-precision identification of cotton leaf diseases in a natural setting. Meanwhile, the proposed bilinear coordinate attention mechanism overcomes the issue that traditional attention mechanisms do not focus on location information and traditional data enhancement techniques tend to introduce noisy information. The proposed bilinear coordinate attention mechanism improves the accuracy of the location of the leaf disease spot and the focus of the feature information and provides an accurate spatial distribution of targets for data enrichment, which boost the model's ability to learn more discriminative features. We describe the methodology of our model and its remarkable effect on a collected cotton leaf diseases dataset. The experimental results show that the accuracy rate of the proposed model is 96.61% and the size is $21.55 \times 10^6$ parameters. This proves that the promising performances of our proposed method compete against eight representative models in the identification of cotton leaf diseases. However, the proposed model also has many limitations. First, the model uses mostly samples of mid- to late-stage disease images and fewer images of early disease. Second, the proposed model shows good performance in the presence of simple backgrounds and better accuracy than traditional models in the presence of complex natural conditions, but further improvement is needed. Finally, more lightweight network models should be considered in order to be more applicable to agricultural production. In future research, more attention will be placed on optimizing the model for performance improvement and weight reduction.

**Author Contributions:** Conceptualization, M.S. and J.Z.; data curation, M.S.; methodology, M.S. and N.Z.; project administration, J.Z.; resources, J.Z.; software, Y.Z.; supervision, J.Z.; validation, N.Z.; writing—original draft preparation, M.S.; writing—review and editing, P.H., S.Z. and J.Z. All authors have read and agreed to the published version of the manuscript.

**Funding:** This work was supported by the National Natural Science Foundation of China (No. 31971792, No. 31901240, No. 32160421), and the Innovation Project of the Chinese Academy of Agricultural Sciences (No. CAAS-ASTIP-2016-AII), and Central Public-interest Scientific Institution Basal Research Fund (No. Y2022QC17, JBYW-AII-2021-08, No. JBYW-AII-2021-29, No. JBYW-AII-2021-34, No. CAAS-ZDRW202107), and Nanfan special project, CAAS, Grant Nos. YBXM10, YDLH01 and YDLH07. Chinese Academy of Agricultural Sciences, Y2022XK24.

**Institutional Review Board Statement:** Not applicable.

**Informed Consent Statement:** Not applicable.

**Conflicts of Interest:** The authors declare no conflict of interest.

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
