# Peer review of "Identification Method of Cotton Leaf Diseases Based on Bilinear Coordinate Attention Enhancement Module"

_agronomy, doi:10.3390/agronomy13010088_

Round 1

Reviewer 1 Report

This paper presents a cotton leaf disease identification model based on bilinear coordinate attention enhancement module on the basis of ResNet34. The results showed that the proposed model overcomes the issue that traditional attention mechanisms do not focus on location information and traditional data enhancement techniques tend to introduce noisy information. This is valuable for related studies and practices. The paper is well structured, with sound descriptions and analysis. The accuracy evaluation of image classification need to improve, because some np-disease regions seemed covered by the hot map. Besides, the reason why the model overcomes shortcomings of others need to analyze.  Detailed comments are listed below.

1、  Line 114-118Suggest put these in the Conclusion part. The objective of this study should be clarified and refined

2、  Line 140Constructed of cotton disease datasetcheck the Language.

3、  Line 174taken as the output The residual, put a comma after ‘comma’.

4、  Line 320-335from1to5, Explain below the table and make comprehensive analysis and comparison in the text.

5、  Line 339Table 4. Explain the meaning of each abbreviation and symbol

6、  Line 341the proposed model can accurately classify.., Figure 6 did’t show the accuracy

7、  Line353-355, from atoc, No need to repeat the same information as the Figure title

8、  Line357the proposed model can accurately locateThe range of heat map is beyond the disease area, so the method of precision evaluation needs to be clear

9、  Line 360-366Figure 7(c) shows shadows seemed covered by the hot map, shadows looks similar to disease regions, how the method differentiate them?

10Why the proposed model better than others? Need to further analyze in the Discussion section.

Reviewer 2 Report

The manuscript “Identification method of cotton leaf diseases based on bilinear coordinate attention enhancement module” addressed the cotton leaf disease identification model based on bilinear coordinate attention enhancement module. The manuscript has several critical issues which should be addressed. My comments are below.

The introduction did not discuss precisely the model used its need and its importance.

Limitations in predictions are so missing. The authors should discuss the major limitation in using their proposed model.

I did not find any substantial novelty in this work. Please highlight it in the introduction.

How will their model be useful in other counties?

What are the experimental area cotton fields in Langfang Scientific Research Pilot Base of Chinese Academy of Agricultural Sciences?

The model is not explained very clearly, how to validate the results obtained by the proposed model?

Some typo needs to be corrected; what does the given example on line 217 denotes? [ , ].

Similarly please see line 237

Spaces between numbers and units are missing

Reviewer 3 Report

Corrections in the body of the attached text.
